# Growth Factor Loaded Thermo-Responsive Injectable Hydrogel for Enhancing Diabetic Wound Healing

**DOI:** 10.3390/gels9010027

**Published:** 2022-12-29

**Authors:** Vyshnavi Tallapaneni, Lavanya Mude, Divya Pamu, Vasanth Raj Palanimuthu, Sai Varshini Magham, Veera Venkata Satyanarayana Reddy Karri, Madhukiran Parvathaneni

**Affiliations:** 1Department of Pharmaceutics, JSS College of Pharmacy, JSS Academy of Higher Education & Research, Ooty 643001, India; 2Department of Pharmaceutical Biotechnology, JSS College of Pharmacy, JSS Academy of Higher Education & Research, Ooty 643001, India; 3Department of Pharmacology, JSS College of Pharmacy, JSS Academy of Higher Education & Research, Ooty 643001, India; 4Department of Biotechnology, Harrisburg University of Science & Technology, 326 Market Street, Harrisburg, PA 17101, USA; 5Arni Medica, 4475 South Clinton Ave, Suite 230, South Plainfield, NJ 07080, USA; 6CRC Pharma LLC, 333 Littleton Road, Parsippany, NJ 07054, USA

**Keywords:** epidermal growth factor, doxycycline, thermos-responsive, injectable hydrogel, diabetic wounds

## Abstract

**Background:** Diabetic wound (DW) is the most devastating complication resulting in significant mortality and morbidity in diabetic patients. The objective of the current study was to formulate Epidermal Growth Factor loaded Chitosan nanoparticle impregnated with thermos-responsive injectable hydrogel with protease inhibitor. EGF, shown in all stages of wound healing from inflammation to proliferation and remodelling, combined with Doxycycline, a well-known anti-inflammatory and anti-bacterial drug, could be a better strategy in diabetic wound healing. However, EGF’s low stability makes it difficult to use. **Methodology:** The nanoparticles were prepared using the ionic gelation method. The prepared nanoparticles were evaluated for particle size, zeta potential, entrapment efficiency, and SEM studies. Further, the optimized nanoparticle batch was loaded into hydrogel with a protease inhibitor. The hydrogel was evaluated for morphology, protease degradation, in vitro drug release, anti-bacterial activity, cell migration, in vitro cell biocompatibility, and in vivo wound healing studies. **Results and Conclusion:** The particle size analysis of nanoparticles revealed the size (203 ± 1.236 nm), Zeta potential (+28.5 ± 1.0 mV), and entrapment efficiency of 83.430 ± 1.8%, respectively. The hydrogel showed good porous morphology, injectability, thermo-responsive, biocompatibility, and controlled drug release. In vitro anti-bacterial studies revealed the potential anti-bacterial activity of doxycycline against various microbes. In vivo data indicated that combining EGF and DOX considerably reduced inflammation time-dependent than single-agent treatment. Furthermore, histological studies corroborated these findings. After topical application of hydrogel, histopathology studies revealed significant collagen synthesis and a fully regenerated epithelial layer and advancement in all three stages (proliferation, remodelling, and maturation), which are required to improve the diabetic wound healing process by any dressing. These findings demonstrated that hydrogel promoted cutaneous wound healing in STZ-induced rats by suppressing inflammation at the wound site. Furthermore, histological studies corroborated these findings. After topical application of hydrogel, histopathology studies revealed significant collagen synthesis, a fully regenerated epithelial layer, and advancement in all three stages (proliferation, remodelling, and maturation), which are required to improve the diabetic wound healing process by any dressing. These findings demonstrated that hydrogel promoted cutaneous wound healing in STZ-induced rats by suppressing inflammation at the wound site.

## 1. Introduction

DWs is a progressive wound on diabetic skin, characterized by abnormally delayed wound healing attributable to compromised biological circumstances and insufficient immunological responses [1]. Despite significant advances in healthcare technology over the past decades, DW continues to place an enormous financial and emotional strain on sufferers. DW is related to a greater menace of limb loss owing to amputation, which decreases survival, on top of the established detriments such as recurrent relapse, protracted treatment, and costly treatment [2]. Clinically, the brutality of DW in ten years of evolution may be castoff as an independent forecaster of death. Current literature has shown that the link between mortality and DW is more vital than that of many CVDs such as peripheral arterial disease, coronary artery disease, and stroke [3,4]. The 5-year existence rate after a new occurrence of DW is approximately 50–60%, according to research by Armstrong et al., making it worse than many malignancies [5]. These factors indicate the critical need for immediate progress toward a more successful therapeutic strategy for DW therapy (i.e., effective DWH) [6].

Hydrogels have long been acknowledged as a valuable tool in treating wounds. Hydrogels prevent subsequent infections by removing excess exudations and foreign substances from the wound surface via their specific porous architecture. The high relative water-content hydrogels may also provide a moist environment around the wound, promoting healing [7]. High user acceptability is a result of these hydrated materials’ non-adherent nature and their capacity to provide warm insulation at the boundary between wound and dressing, which can significantly lessen the discomfort experienced by patients [8]. Hydrogels can be made from various components, including natural and synthetic polymers, lipids, and surfactants [9]. Biocompatible, biodegradable, cheap cytotoxicity, antimicrobial, hemostatic, and anticancerogenic properties are vital for its role as a wound dressing. The polymer, i.e., chitosan, an N-acetyl derivative of chitin obtained by N-deacetylation, has gained significant attention due to these qualities [10].

Many studies have been reported recently on hydrogel-based dressings in treatment of normal and diabetic wounds. Ma et al., 2022 have reviewed about the effects of photo-crosslinking hydrogels in promoting wound healing. Due to the highly alike biomechanical characteristics to the ECM of photo-crosslinking hydrogel, macromolecular hydrogel bone materials along with the loaded drugs and bioactive substances can maximize the role of promoting wound healing, such as antibacterial, anti-inflammatory, antioxidation, hemostasis, tissue remodelling, angiogenesis, inhibition of scar hyperplasia, and moisturizing. However, there are limited studies carried out in humans and clinical studies have to be carried out to know the complications associated and advantages of the same [2].

Ren et al., 2021 have prepared a series of injectable and antioxidative HT/QGA hydrogels based on hyaluronic acid-graft-tyramine (HT) and quaternized chitosan-graft-gallic acid (QGA) through the cross-linking of catechol groups with HRP/H_2_O_2_ as the initiator system. They also stated that HT_1_/QGA_0.3_ hydrogel demonstrated a faster rate of wound closure than the Control and TM Film groups, suggesting that HT_1_/QGA_0.3_ hydrogel can significantly promote wound healing [3]. 

However, all these methods only focused on improving the delivery of growth factors and neglecting their degradation in the proteases environment in chronic wounds. Hence, future studies should address both inflammation, infection as well tissue regeneration aiming to maximize diabetic wound healing.

Hydrogels with incorporated growth factors and cytokines, such as vascular endothelial growth factor (VEGF), insulin-like growth factor (IGF), epidermal growth factor (EGF), transforming growth factor (TGF-β), fibroblast growth factor (FGF), platelet-derived growth factor (PDGF), and tumour necrosis factor-α (TNF-α), are commonly used to improve the efficiency of wound repair [11]. EGF is a polypeptide with 53 amino acids that acts as a growth factor. Because of its ability to increase significantly keratinocyte and fibroblast proliferation and motility, granulation tissue formation, and extracellular matrix (ECM) creation, it has been established that it plays a crucial role in wound healing. Nevertheless, hydrogels have outstanding physicochemical features, their loading capacity of proteins/polypeptides, such as EGF, is low [12]. In addition, the high-water content of hydrogels can lead to adverse effects for patients attributed to an overdose of an agent, such as EGF-induced malignancies [13], and 2) the degradation of therapeutics, as DFU/chronic wounds typically show increased protease action, diminished cellular activity, and an exaggerated level of pro-inflammatory cytokines, all of which are predominantly unfavourable for EGF. Hydrogel-mediated EGF therapy for DWH is greatly impeded by unfavorable conditions [14].

However, because of the persistent presence and co-location of bacteria, sloughs, and biofilms in chronic wounds like DFUs, these wounds are especially vulnerable to infection [15]. Even though some hydrogel materials, such as chitosan, have bacteriostatic qualities, it is common practice to introduce bacterial inhibitors into hydrogels to boost their antimicrobial effectiveness. The tetracycline antibiotic doxycycline hyclate (DOX) inhibits matrix metalloproteinases. When bound to the 30S ribosomal subunit, it blocks the production of proteins in bacteria. DOX exhibits bacteriostatic action in vitro and clinical infections against many different types of bacteria. DOX is also an effective treatment for MRSA infections [16].

Since nanocarriers may (1) increase the bioavailability, stability, and security of payloads (such as EGF) and (2) permit a sustained and controlled release of drug molecules inside the polymeric matrix, the nanoencapsulation technique may provide a practical means for administering EGF in the hydrogel system [17].

Our goal in this research was to develop and test a thermo-responsive injectable hydrogel containing DOX and EGF nanoparticles (C-EGF-D IHG) for application in the DWH. Before being released onto the wound surface, DOX prevents the proteases. The EGF molecules were protected from degradation and released in a controlled manner by being enclosed in a chitosan-polyanions cross-linking nano-device. hypothesisedzed that the C-EGF-D IHG we created, with its persistent anti-bacterial and cell growth promotion potential, would aid in diabetic/chronic wound healing.

## 2. Results and Discussion

### 2.1. Characterization of EGF-CS NPs

Formulation of blank nanoparticles and EGF-CS NPs have been prepared using the ionic gelation method, and the formulation parameters have been tabulated in Table 1. 

The ZP and PS of the freshly synthesised CS-NPs and EGF-CS NPs were determined (also agree are in good agreement with the predictions made by the researchers.

The PDI is a critical parameter that measures the width of a particle’s size distribution in a sample. The ideal PDI value should be as near to 0 as possible (0 being for monodispersed particles). Low PDI values also imply that the dispersion is relatively homogeneous.

The zeta potentials of the prepared CSNPs were found to have +21.3 mV and drug loaded NPs was found to be +28.5 mV (Figure 1 and Figure 2). Because of the protonation of free amino groups, CS had a positive amount, and the CSNPs had the same charge. The zeta potential must be above ±30 mV to produce a stable nanosuspension, regardless of the charge. The encapsulation efficiency of EGF-CSNPs was found to be 83.430 ± 0.397 percent in our investigation. The benefit of improved encapsulation efficiency is that a more significant molecule is deposited at the target site, extending its stay there. Several variables contribute to the EGF-CSNPs excellent encapsulation effectiveness. First, a higher amount of CS has a more remarkable ability to form an ionic gel, which prevents EGF from moving to the external phase and boosts drug encapsulation efficiency, resulting in increased drug loading. Second, the vast number of amine groups in the CS enhances electrostatic contact between the cationic and amino groups. The drug loading was found to be 19.76 ± 1.73% which depicts that 1 mg of nanoparticle contain 0.19 mg of drug. The particle size of blank and drug loaded nanoparticles is found to be 168.85 & 201.3 nm respectively Figure 3 and Figure 4.

In conclusion, a less than 200 nm particle size with good entrapment effectiveness offers better drug release control and interacts with cells (cell infiltration). A Zeta potential of near or more than + 30 mV is required for good stability. EGF-CSNPS also has a positive zeta potential, which aids in hydrogel adhesion to negatively charged biological membranes. The EGF-CSNPs from formulation three have been chosen as a reference batch for future testing in light of the preceding.

The SEM analysis was performed to study the morphological characteristics of the formulated NPs and found to be a spherical shape with uniform distribution (Figure 5).

### 2.2. Preparation and Characterization of C-EGF-D IHG

Dressings are essential for ulcer treatment among many wound-healing management [4,5]. An ideal wound dressing would fulfil the following criteria: (1) create a moist wound-healing environment to promote the proliferation and migration of keratinocytes and fibroblasts and enhance collagen synthesis [6,7]; (2) allow for gas exchange, and thermal insulation; (3) be biocompatible, nontoxic, and non-allergenic; (4) prevent secondary infections from developing; and (5) be easily removed without causing further damage to the wound [8].

Three-dimensional bioengineered substituents such as scaffolds, nanofibers, films, microfibers, and hydrogel have received attention in this scenario because of their multiple impacts and unique qualities, namely their likeness to original tissue [9]. Hydrogels are in high demand among bioengineered replacements because they can keep wounds wet and absorb tissue exudates with a high oxygen supply to stimulate tissue regeneration [10].

Wichtlerle and Lim pioneered for the first time the use of poly(2-hydroxyethyl methacrylate) (PHEMA)-based hydrophilic gel as a biomaterial in contact lens applications in 1960 [11]. Hydrophilic gels have been widely used in biomedical applications such as tissue engineering, drug delivery systems, contact lenses, wound treatment, etc. Vann Bemmelen coined the phrase hydrophilic gel as a hydrogel for the first time in 1984 [12]. 

Hydrogels are polymeric networks that are cross-linked by one or more monomers. It can retain a considerable amount of water or biological fluids within its structures while remaining insoluble. Because of its cross-linking structure resembles biological tissue (ECM, extracellular matrix) [13]. Although hydrogels offer many advantages, they have drawbacks, such as limited mechanical strength, inadequate wound filling, and difficulty encapsulating medications or bioactive ingredients [14].

Based on these constraints, some researchers have created in situ gelling injectable hydrogels for easy encapsulation of therapeutic drugs, complete filling of the wound region, and a simple, minimally invasive surgical technique [15]. Elisseeff et al. (1999) reported using in situ hydrogels for tissue regeneration [16]. In situ gel formation occurs in a polymeric solution; however, the polymer solution undergoes physical and chemical cross-linking following injection and to form a gel and provide a suitable coupling between g el and host tissues. More crucially, in situ gelling injectable hydrogels enable the delivery of cells and growth factors locally, leading to faster and more complete skin regeneration [16].

When growth factors are introduced into biological fluids, they typically have a limited half-life and are swiftly destroyed. As a result, it is critical to establish a practical mechanism for delivering growth factors to maintain their therapeutic efficacy. Controlled delivery techniques incorporating growth factors into polymeric biomaterials have been developed to extend exposure and reserve the stability of growth factors. EGF has been shown to stimulate in vitro cell proliferation and in vivo wound healing, among the growth factors. EGF delivery that is localized and prolonged reduces the size of the wound area dramatically. It has also been noted that when applied to an injured location, numerous proteases can quickly degrade EGF. On the other hand, carbohydrate portions of natural glycoproteins shield polypeptide chains against proteolysis. As a result, protecting EGF from protease degradation through conjugating low molecular weight sugars like low molecular weight chitosan, dextran or dextrin could be one of the most capable approaches for increasing its in vivo efficacy.

So far, a wide range of natural or synthetic polymers has commonly been used in producing hydrogels. Chitosan, an N-acetyl derivative of chitin obtained through N-deacetylation, has received much attention due to favourable biological properties such as biocompatibility and biodegradability, low cytotoxicity, antimicrobial, hemostatic, and anticancerogenic, which are essential for its use as a wound dressing. Chemically, CS is a weak basic soluble in pH 6.5 solvents but insoluble in non-polar solvents. CS’s two-dimensional chemical structure contains two reactive functional groups, amino (-NH2) and hydroxyl (the presence of free amino groups, the CS is protonated in acidic liquids and solubilized to aid in creating a bio-adhesive film by air drying. The microstructure of the hydrogels was observed using FESEM and found to have a three-dimensional pore structure (Figure 6). This three-dimensional porous structure would benefit during the initial phase of wound healing, i.e., homeostasis to absorb interstitial fluid or blood. Moreover, it may also allow the transport of cellular nutrients and promote wound healing.

The initial gel temperature of the C-EGF-D IHG was 37 °C, while the initial gel temperature declined to around 32.5 °C, with a much greater storage modulus G′. This suggests that adding GP to the CS gel system enhances the gel’s mechanical qualities and speeds up the gel’s speed. Furthermore, the higher the GP concentration in the CS/GP system, the lower the initial gel temperature and the more significant the mechanical property (Figure 7). C-EGF-D IHG system has a shear thinning characteristics at room temperatures 25 °C and 37 °C, thereby the feature of injectability.

Drug release from both the formulations EGF-CSNPS and C-EGF-D IHG was carried out to evaluate the degree of drug release regulated by the nanoparticles (EGF-CSNPS) and hydrogel (C-EGF-D IHG). In vitro release of EGF from CSNPs and C-EGF-D IHG in SWF, pH 7.4 at 37 °C, is shown in Figure 8. The release of EGF from the CSNPs demonstrated a biphasic trend characterized by quick release for 24 h accompanied by a sustained release phase. In the initial 24 h, 22.03 ± 0.28% of the drug became available from the CSNPs, while C-EGF-D IHG merely released 13.59 ± 0.32% of the drug. The C-EGF-D IHG delayed the initial drug release, which can be an outcome of the time taken to maintain the drug in contact with the receptor under the initial condition. After 24 h of study, 44.01 ± 0.089% from CSNPs and more than 26.43 ± 0.47% from C-EGF-D IHG of EGF were released. At 72 h of evaluation, 67.57 ± 0.38% and 51.33 ± 0.05% of the drug were available from CSNPs and C-EGF-D IHG. It will be favourable only if the delivery system (C-EGF-D IHG) sustains the drug’s release during the treatment duration. The drug release of 77.56 ± 0.32% from the C-EGF-D IHG at 120 h will ensure effective drug release over five days and above for the treatment. Controlled drug delivery mediated by C-EGF-D IHG can be beneficial in minimizing the inflammation for an extended time and the rate of reapplication that pays to current wound healing and treatment. In contrast, more than 90% of DOX was released at 24 h from C-EGF-D-IHG. This initial burst release of DOX is required to control the infection load and to employ abrupt chemoprophylaxis on the wound area.

On average, chronic wounds have proteases of 87 ug/mL. Hence, 50 ug/mL was selected for the savinase enzyme test in this study. It was confirmed that adding DOX and EGF-CS NPs to hydrogel significantly increased its stability towards proteolytic degradation of EGF compared to free EGF and EGF-CS NPs (Figure 9).

### 2.3. In Vitro Evaluation

#### 2.3.1. Analysis of Cytotoxicity Studies

In this in vitro cytotoxicity study, the 3T3-L1 fibroblast cell lines were used as the test subjects. Fibroblasts have a vital role in producing growth factors, the formation of extracellular matrix, the production of collagen, and the stimulation of keratinocytes. These components all contribute to forming the structural framework of skin tissue is necessary for wound healing. The cytotoxicity testing revealed that the formulation under investigation had no adverse effects on the cell lines tested. Compared to the control group, the C-EGF-D IHG produced significant cell proliferation, which the EGF-CSNPs followed. The results indicate that the C-EGF-D IHG was biocompatible, as evidenced by the constant growth of fibroblasts. The presence of biopolymer (CS) was responsible for the increased cell proliferation observed in both participants. A significant change was observed in the C-EGF-D IHG group, which could be attributable to EGF’s mitogenic activity associated with the phosphorylation of P42/44 MAP kinases (MAPKs). We have confirmed that EGF has mitogenic effects on NIH 3T3 fibroblasts, conjunctival fibroblasts, and airway epithelial cells. The biopolymer is also cell-friendly and biocompatible, and CS allows favourable cell growth (Figure 10).

#### 2.3.2. Anti-Bacterial Activity

If left untreated, the DW becomes infected and spreads to deeper layers of skin tissue, eventually leading to gangrene and amputation. Even though DW infection is polymicrobial, bacteria such as *E. coli*, *P. aeruginosa*, *MRSA*, *and S. aureus* are commonly detected in DW and account for around 70% of mixed infections (Table 2). Anti-bacterial tests were thus carried out against these four microorganisms to investigate the effect of CS and DOX-CS-IHG on them. At 1 ug/mL, DOX-CS-IHG demonstrated substantial anti-bacterial activity against *MRSA and S. aureus* compared to *E. coli and P. aeruginosa*. This could be due to DOX’s anti-bacterial effect against Gram-positive bacteria (*S. aureus and MRSA*) instead of Gram-negative bacteria (*E. coli and P. aeruginosa*) Figure 11 & Table 3.

#### 2.3.3. Cell Migration Assay

The ability of EGF-CS-NPs and C-EGF-D IHG to produce in vitro wound closure was accustomed to study the impact of C-EGF-D IHG and EGF-CS-NPs loaded hydrogel on the migratory capability of 3T3-L1 fibroblast cells. Over 24 h, the rate of wound closure was monitored. Due to the brief duration of the experiment, cell proliferation may have shown a minimal effect on wound closure rate. Cells treated with C-EGF-D IHG migrated faster than cells treated with EGF-CS-NPs alone. C-EGF-D IHG improved in vitro wound closure 24 h after wounding compared to EGF-CS-NPs and control (Figure 12). The increased migration rate of fibroblasts in the C-EGF-D IHG treated group might be due to the chemoattractant property of EGF towards fibroblasts, leukocytes, and keratinocytes as reported by Lee et al. [17].

### 2.4. In Vivo Evaluation

#### 2.4.1. Wound Healing Activity

The STZ-induced rodent model was used to study DW healing. From the third day of STZ treatment, the typical diabetes indications of elevated blood glucose, polyphagia, polydipsia, polyuria, and weight loss were detected. A minimum glucose threshold of 300 mg/dL was maintained to ensure homogeneity in the trial. On days 0, 7, 14, and 21, the wound contraction in each group was measured using a grid approach. With a *p* < 0.001 significance level, the findings are statistically significant. On day 7, diabetic rats treated with C-EGF-D IHG wound repair were substantially faster than EGF-CS NPs, placebo, and untreated rats, with 52.282.96 percent compared to 72.512 percent for placebo versus 94.282 percent for untreated.

Furthermore, the C-EGF-D IHG minimized inflammation and avoided ulcer formation, whereas other groups experienced incorrect granulation tissue production and ulceration (Figure 13 and Figure 14). The collagen that was deposited was compact and densely aligned. The increased wound closure in this study was consistent with that seen in other studies using EGF and biopolymer treatments.

#### 2.4.2. Histopathology Analysis

Hydrogels are also well-liked by patients due to their stated properties like flexibility, nonadherence, and keeping the wound’s surface moist, which may result in a noticeable reduction in discomfort. Various studies have demonstrated that wound healing can be delayed and impeded in chronic wounds, including DWs. In vivo findings revealed that wound closure (percentage) was much higher in the C-EGF-D IHG group. Even though the EGF-CS NPs groups were comparable to the produced formulation, the C-EGF-D IHG can play a superior function in wound healing because of its prolonged release and delayed proteolytic destruction.

Figure 15 and Figure 16 depicts demonstrative wound histopathology for all the treated groups. Many blood vessels upright to the wound area are one of the granulation tissue indicators. The granulation tissue replaced with fibrous tissue is an essential trait in favour of healing, commencing re-epithelialization, and forming the cutaneous adnexa. Reepithelialization and wound closure are then necessary to complete the healing process.

Our histological findings showed that re-epithelialization was much more in formulation-treated groups than in control.

On days 7, 14, and 21, the C-EGF-D IHG-treated group saw faster re-epithelialization, fibroblast, and keratinocyte migration, resulting in a thicker neo-epidermal layer visible on tissue slices. Our findings in the in vitro scratch experiment corroborated these findings. Collagen deposition also shows fibroblast activity in newly produced granulation tissue, which is critical in the proliferative and remodelling phases of the healing process. EGF activates the protein kinase C pathway, which increases fibroblast migration. EGF also can cause keratinocyte proliferation and migration. Due to the enormous modulatory activity of both EGF and CS in wound healing, C-EGF-D IHG treated groups achieved significant re-epithelialization and collagen deposition. Furthermore, CS gradually depolymerizes into N-acetyl glucosamine (NAG), which stimulates fibroblast proliferation and aids in the orderly deposit of COL. The fast release of DOX from the C-EGF-D IHG afforded protease inhibitory activity, allowing EGF to operate and anti-inflammatory and anti-bacterial effects on the DW.

Our in vivo data showed that the prepared dressing could intensify the healing rate pointedly. Its properties like enhanced autolytic debridement, nonreactive with biological tissue and rehydrating of dead tissues make them more suitable as wound dressings.

#### 2.4.3. MMP-9 Estimation

Figure 17 shows the MMP-9 estimation data, showing that the C-EGF-D IHG-treated group had significantly lower MMP-9 expression than the EGF-CS NPs, placebo, and control-treated groups. MMP-9 levels were significantly lower in the C-EGF-D IHG (14.52 ± 0.52 ug/mL) treated group on day seven after wounding compared to the EGF-CS NPs (17.97 ± 0.75), placebo (21.52 ± 0.92) and control (24.31 ± 0.84) treated groups. On day 14, a comparable decline was also noticed. On day 21, MMP-9 levels in the C-EGF-D IHG (3.40 ± 0.83) treated group was significantly lower than in the EGF-CS NPs (11.453 ± 1.1), placebo (15.243 ± 0.42) and control (19.303 ± 0.23) treated groups. This emphasizes the significance of DOX in boosting DW repair by lowering MMP-9 levels. These findings align with a previous study by Cui et al., who found that DOX treatment reduced MMP-9 levels in periodontitis mice.

#### 2.4.4. TNF-α and IL-10 Determination

In the inflammatory phase of DW, both TNF-α and IL-10 levels in wound fluid are essential. Figure 18 and Figure 19 depict the findings of the TNF-α and IL-10 determinations. Up to 7 days after wounding, TNF-α levels were higher in the control and placebo groups than in the EGF-CS NPS and C-EGF-D IHG treated groups. When compared to placebo (1657.86 ± 76.26 pg/mL) and control (1865.2266 ± 72.244 pg/mL), TNF-α levels were significantly lower in C-EGF-D IHG (804.863 ± 57.37 pg/mL), and EGF-CS NPs (1387.523 ± 63.248) treated groups on day 14. In comparison to the placebo (1548.293 ± 77.1087 pg/mL) and control (1842.7733 ± 67.377 pg/mL) groups, the C-EGF-D IHG (356.42 ± 78.316) and EGF-CS NPs (1001.813 ± 53.273) treated group showed a significant drop in TNF-α levels on day 21.

Furthermore, IL-10 production was reduced in the placebo and control groups 21 days after injury. While C-EGF-D IHG groups produced more IL-10, the control groups did not. On day seven after wounding, C-EGF-D IHG (897.27 ± 0.987) treated groups had significantly higher IL-10 levels than EGF-CS NPs (625.833 ± 1.752), placebo (281.683 ± 0.958), and control (212.1566 ± 1.289) treated groups. On day 14, similar results were reported. When comparing the C-EGF-D IHG (1365.44 ± 0.34) to the EGF-CS NPs (1044.71 ± 0.932), placebo (427.8566 ± 1.542) and control (325.74 ± 1.486) groups on day 21, the IL-10 levels were significantly higher in the C-EGF-D IHG. 

TNF-protein expression was considerably reduced on all days, whereas IL-10 was proportionately enhanced in the C-EGF-D IHG-treated groups related to the control and placebo-treated groups. These findings are encouraging since they suggest that C-EGF-D IHG may have anti-inflammatory properties.

C-EGF-D IHG-treated groups experienced a reduction in inflammatory mediators (MMP-9, TNF-α, and IL-10) to the levels observed in placebo and control-treated groups. This diabetic pro-inflammatory reversal by C-EGF-D IHG may be explained by decreased recruitment and activation of inflammatory mediators, eventually improved inflammatory phase resolution, which finally facilitated the progression to the subsequent wound healing phases. 

However, due to the complicated pathophysiology of DW, these treatments are not always effective and cost a lot of money. As a result, developing a single therapeutic strategy (multi-mechanism-based products) to close all the gaps could be advantageous in treating DW.

## 3. Conclusions

DWs are the leading cause of death among diabetic individuals. DW care is centred on early detection, patient education, and improved therapeutics. Even though numerous papers have been published and suitable substances are currently being evaluated in clinical trials for therapy, a conclusive body of data has yet to be produced. With a solid understanding of DW pathophysiology, aetiology, and microbiological aspects, more sophisticated wound dressings that do more than “cover & conceal” could be developed. These characteristics would be able to address the most pressing issues in non-healing wounds while prioritizing healing. Although the pathophysiology of DWs is complex, persistent inflammation, infections, and a lack of tissue regeneration (tissue management) hinder DWH, leaving wounds in a chronic non-healing state.

Therapeutic dressings, on the other hand, have been created to ensure proper distribution of antimicrobials [18,19] or to process changes in the inflammation pathway at the wound site. However, none of the studies found any dressings that adequately addressed infection, inflammation, and viable tissue management. EGF, shown in all stages of wound healing from inflammation to proliferation and remodelling, combined with DOX, a well-known anti-inflammatory and anti-bacterial drug, could be a better approach in DWH. However, EGF’s low stability makes it difficult to use. As a result, a novel injectable hydrogel was created in this study by integrating EGF into CSNPs to expand their stability, followed by impregnation of produced EGF-CS NPs and DOX into CS hydrogel (C-EGF-D IHG) for protecting against protease degradation and enhanced tissue regeneration.

The formulated EGF-CS NPs were evaluated for particle size, zeta potential, SEM, and the C-EGF-D IHG for morphology, biodegradability, biocompatibility, in vitro drug release, and in vivo wound healing studies in STZ-induced diabetic rats. The EGF-CSNPs have been successfully prepared using the ionic gelation method with particle size, zeta potential, and EE of 201.3 nm, +28.5 mV, and 83.430 ± 1.8%, respectively. The C-EGF-D IHG has been successfully prepared by adding EGF-CSNPs to the DOX-loaded CS-β-GP solution. The C-EGF-D IHG showed good porous morphology, injectability, thermo-reversible biocompatibility, and controlled drug release. In vitro, anti-bacterial studies revealed the potential anti-bacterial activity of DOX against various microbes present in DWs. These results also added that the combination of CS and DOX has a synergistic anti-bacterial effect over individual treatments. In the current era of antibiotic resistance, increased resistance of microorganisms to antibiotics has led to severe challenges in treating infectious diseases. Hence, there is a need to develop alternative natural or combinational therapies. 

In vivo data revealed that combining EGF and DOX considerably reduced inflammation time-dependent than single-agent treatment. Furthermore, histological studies corroborated these findings. After topical application of C-EGF-D IHG, histopathology studies revealed significant fully regenerated epithelial layer, collagen synthesis, and advancement in all three stages (proliferation, maturation, and remodelling), which are required to improve the DWH process by any dressing. These findings demonstrated that C-EGF-D IHG promoted cutaneous wound healing in STZ-induced rats by suppressing inflammation at the wound site.

Finally, our findings suggest that combining EGF and DOX had a superior anti-bacterial (in vitro) and anti-inflammatory effect than monotherapy, as evidenced by reduced inflammation and improved wound contraction in C-EGF-D IHG treated groups. We believe both medications’ potent anti-inflammatory properties boosted each other’s response. Their synergistic effects on inflammation and infection suggest that EGF and DOX can be utilized as adjuvants in treating DWs with infections.

Regarding biocompatibility, porosity, biodegradation, controlled release, proliferation, anti-inflammatory, and anti-bacterial properties, the novel C-EGF-D IHG met the properties of an ideal DW dressing: critical for tissue regeneration in DWs. As a result, the current study suggests combining EGF, DOX (anti-inflammatory and anti-bacterial), and CS (drug carrier, wound healing, and biomaterial for regenerative medicine) is a promising approach for addressing several pathological manifestations of DWs and improving wound healing competence.

This section is not mandatory but can be added to the manuscript if the discussion is unusually long or complex.

## 4. Materials and Methods

### 4.1. Preparation and Characterization of EGF-Loaded Chitosan Nanoparticles (EGF-CSNPs) 

Low-molecular-weight chitosan (600 mg) was dissolved in acetic acid (100 mL, 1 percent *w/v*), and the pH of the solution was adjusted to 5.5 using sodium hydroxide (NaOH) (1 M). Human EGF (300 μgmL^−1^) was dissolved in the chitosan solution to yield a protein concentration of 50 μgmL^−1^, which was then used to make the gel. Following that, sodium tripolyphosphate (15 mL, 0.1 percent (*w*/*v*)) was added dropwise to the aforesaid EGF-chitosan solution while maintaining steady stirring, followed by an extra step of agitation (2000 rpm) at room temperature for 4 h to complete the experiment. The generated EGF-CSNPs were then used for additional characterization [20].

#### 4.1.1. Particle Size Determination and Zeta Potential 

In distilled water, EGF-CSNPs (100 µgmL^−1^) were suspended. Each measurement consisted of 60 accumulations repeated in triplicate before being averaged to get the standard deviation error. The diluted suspension for surface charge measurement filled the zeta quartz cell. The zeta potential was acquired in triplicate after determining the optimum density for starting the test [20].

#### 4.1.2. Entrapment Efficiency (EE)

EE was measured indirectly by measuring the amount of free rhEGF (non-encapsulated) retrieved via centrifugation (15,000 rpm; 4 ℃; Remi equipment’s Ltd., India). After that, each sample was diluted at 1:10,000 in DPBS solution. The free rhEGF was calculated using the manufacturer’s instructions and a commercially available Sandwich Enzyme-Linked Immuno-sorbent Assay kit for human EGF (human EGF ELISA development kit, PeproTech, Cranbury, NJ, USA) [21]. The following equation was used to calculate the encapsulation efficiency:

All tests were performed in triplicate, and the results are reported as the means ± SD.
EE%=Initial amount of rhEGF−Free rhEGF/Total quantity ofEGF×100

#### 4.1.3. Morphological Characterization

The morphology of EGF-CSNPs was determined using scanning electron microscopy (SEM) (SEM, JEOL 6500F). Samples were mounted on aluminium SEM stubs using a double-sided carbon sticker. Samples were placed in the coating crucible, flushed with argon, and then coated with gold to prepare them for SEM imaging [22].

### 4.2. Preparation and Characterization of Chitosan-EGF- Doxycycline Loaded Thermo-Responsive Injectable Hydrogel (C-EGF-D IHG)

For 4 h, chitosan (2%) was dissolved in an acetic acid solution (0.1 M, 100 mL) with constant stirring. EGF-CSNPs (20 ugmL^−1^) and DOX (0.5 percent *w*/*v*) were added to the prepared solution. A 50 percent aqueous solution of beta glycerophosphate was carefully added drop by drop to the above under continuous stirring on ice to achieve a precise, and homogeneous liquid solution in a final volume of 5 mL. The hydrogel was created by heating a chitosan/glycerol phosphate solution to 37 degrees Celsius.

#### 4.2.1. Morphology

The cross-sectional morphology of hydrogel was determined by inserting the freeze-dried samples into the coating crucible, flushed with argon, and coated with gold making them ready for SEM imaging [22]. 

#### 4.2.2. Rheological Characterization and Injectability

Two rotational rheometer modes described the thermo-sensitive gel: temperature and viscosity. The temperature rose at a rate of 0.5 °C/min from 25 °C to 45 °C in the temperature mode. The viscosity model set the shear rate between 0.1 s^−1^ to 400 s^−1^ and the temperature between 25 and 37 degrees Celsius. The remaining parameters are set to their default values. The thermo-sensitive sol-gel transition behaviour was explored using a temperature cycle step test between 25 °C and 45 °C, with heating and cooling rates of 0.5 °C min^−1^.

The injectability of the C-EGF-D IHG was studied in vitro. C-EGF-D IHG was filled in the 22-gauge syringe and then injected into PBS. The process was photographed.

#### 4.2.3. In Vitro Drug Release Studies

The C-EGF-D IHG was placed on a metal mesh that served as a bed for the experiment. The medium used was phosphate buffer (pH 7.4), gently agitated while maintaining the temperature at 37 degrees Celsius. The EGF-CS NPs and C-EGF-D IHG were first dispersed in phosphate buffer (pH 7.4) for 24 h before the experiment to ensure complete dispersion of the EGF-CS NPs and C-EGF-D IHG.

The diffused samples were obtained at various intervals: 0, 2, 4, 8, 12, 24, 48, 96, and 120 h after the initial collection. According to the results, the concentration of EGF was determined using an EGF-ELISA kit and a colourimetric plate reader. The release of DOX was observed using Ultra-Fast Liquid Chromatography.

### 4.3. Characterization of In Vitro Degradation of Formulations

The EGF-CSNPs and lyophilized C-EGF-D IHG were transferred in 15 mL test tubes. Then, these tubes were filled with 10 mL of PBS (pH = 7.4) containing lysozyme (2 × 104 U/mL) and stored in a 37 °C water bath shaker. The hydrogel was removed from the medium and then freeze-dried for two days at predetermined intervals. 

The degradation rate was calculated by the following formula [23].
Degradation rate (t)% = (W_0_ − W_t_)/W_0_ × 100%, 

W_0_: initial gel dry weight, W_t_: degradation of the dry weight of the gel at time t.

### 4.4. In Vitro Analysis

#### 4.4.1. Cytotoxicity Study

The MTT assay was used to identify a dose for further analysis of plant extracts/drugs for their activity before further testing. In this study, which is one of the cytotoxicity investigations, the assay method 3-(4,5-Dimethylthiazol-2-yl)-2,5-diphenyltetrazolium bromide (MTT) was used to evaluate cell viability using the MTT. For this experiment, the cells were seeded in 96-well plates and allowed to attach to the plate for 24 h. For the next 20 h, 100 uL of different concentrations of 15.5 uM/mL to 500 uM/mL (in increasing order of concentration) of placebo and C-EGF-D IHG were added to the mixture. The cell medium was then replaced with new medium/wells containing MTT, and the cells were cultured for another 4 h in the darkened cells incubator. The supernatant is drained, and 100 µL of DMSO is added to each well of the plate. Detection of absorbance was done at 570 nm using a microtiter plate reader. The blank wells were two per plate devoid of cells [4].
Cell viaility=(Sample absorbance−Blank absorbanc)/(Control absorance−Blank absorbance)×100

#### 4.4.2. Anti-Bacterial Activity

The anti-bacterial activity of the DOX loaded and placebo gels are tested against *S. aureus*, *P. aeruginosa*, *MRSA*, and *E. coli*. Suspensions of the organisms, as mentioned earlier, are prepared from fresh colonies after overnight incubation, and the turbidity is corrected to 0.5 McFarland standards (~105c.f. u/mL). The anti-bacterial activity of prepared injectable gels against these microbes is estimated by the Kirby-Bauer disc diffusion method. Injectable hydrogel samples (100 μL) are sterilized & incorporated into the study. A nutrient agar plate with evenly spread bacterial suspension had the samples placed upon its surface & subjected to overnight incubation (37 °C) followed by zones of inhibition measurement. The clear zone diameter of the bacterial inhibition zone was correlated to antibiotic activity (DOX and DOX-loaded formulation) for Gram-positive and Gram-negative bacteria, respectively [23].

#### 4.4.3. Cell Migration

The scratch assay was carried out using the 3T3-L1 fibroblast cell line (purchased from NCCS Pune, India), which was previously described. A 12-well culture plate with 1–2 mL of warmed medium applied to each well was used for this experiment. After 24 h of growth, the cells were seeded into a 12-well tissue culture plate at a density that resulted in 70–80% confluence after 24 h of growth. Each well of a 12-well culture plate has a growing area of approximately 4 cm^2^ per well. For fibroblasts, it is recommended that 200k cells be plated in each well of a 12-well plate to achieve confluence by the next day (50k cells per cm^2^). Using a 1 mm pipette tip, scraping of the cell layer was done in a straight line once the cells had reached confluence (typically 18–24 h). Using another vertical line to the initial line, scratched across each of the four wells. When generating a scratch, the tip must be kept in contact with the bottom of the well to remove the cell layer.

Nevertheless, excessive pressure should not be used. Immediately after scratching, we gently washed the cell monolayer to remove any detached cells before replacing them with the new medium. Images were taken using a phase-contrast microscope at magnifications of 4× and 10×. Placed in an incubator and imaged every 4–8 h with a phase-contrast microscope until the cells migrated to the chamber’s centre (24–48 h) [4].

### 4.5. In Vivo Wound Healing Studies

#### 4.5.1. Induction of Diabetes

The study employed healthy adult Wistar albino rats (males) weighing roughly 180–200 g. To induce type 2 diabetes, intraperitoneal injection of streptozotocin (70 mg/kg) in cold citrate buffer (0.1 M) with a pH of 4.5 and a high-fat meal (20% proteins, 22% fat, 48% carbs with a total calorific value of 44.3 kJ/kg) were utilized. The animals were housed in ordinary polycarbonate cages with climate control (12:24 h, dark cycle). Rats with blood glucose levels of 250 mg/dL (Glucometer, Accucheck, USA) were separated and monitored for another seven days. Only rats with consistently constant blood glucose levels are evaluated for the investigation. 

#### 4.5.2. Induction of Excision Wound

Under anaesthetic conditions, an open wound (2 × 2 with a depth of 2 mm) was made on the diabetic rat’s depilated dorsal thoracic area (Diethyl ether). The animals were thoroughly monitored once they were sedated. All animal investigations were authorized by the JSS College of Pharmacy’s Institutional Animal Ethical Committee, with protocol number JSSCP/OT/IAEC/Ph.D./02/2021–22. 

#### 4.5.3. Treatment of Diabetic Excision Wound

The rats considered for the study were segregated into four groups (n = 6) and housed individually after anaesthesia recovery. Wounds in Group 1 were untreated but covered with sterile gauze (Control); Group 2 was treated with CS-IHG (Placebo), and Group 3 was treated with EGF-CS NPs (Test formulation) and Group 4 with C-EGF-D IHG, respectively, throughout the study duration [24]. 

#### 4.5.4. Measurement of the Wound Healing

The wound tracing and graphical approach were used to calculate the reduction in mean wound area. The wounds were traced on a sterile translucent OHP sheet on days 0, 7, 14, and 21. The outlined wound regions were calculated using the graph sheet. The reduction in mean wound area was determined using the formula below.
$$Reduction\;in\;mean\;wound\;area=\frac{Wound\;area\;on\;intial\;day-Wound\;area\;on\;particular\;day}{Wound\;area\;on\;intial\;day}\times 100$$

### 4.6. Ex Vivo Studies

#### 4.6.1. Histopathology Studies

On days 7, 14, and 21, each group’s animals were euthanized, and wound tissues were separated and preserved in a 10% formalin solution made in the neutral buffer. Using a microtome, tissues were sectioned at a thickness of 6 m (Model No: RM2135; Leica, United Kingdom). Eosin, hematoxylin, and Masson’s trichome were used to stain the sections and then placed on a glass slide. The photographs were taken at a magnification of 40 using a fluorescence microscope equipped with a 20 MP camera (Model No: MLX-iPlus; Magnus, India). 

#### 4.6.2. MMP-9 Estimation

An enzyme-linked immunosorbent assay kit was used to determine how much MMP-9 was in the wound tissue cut out. The method used was in line with the instructions given by the manufacturer. The tissues that had been cut were homogenized and centrifuged. The supernatant was diluted 100-fold with the assay buffer. Using a microplate reader, Tecan-i-control, we found out how much MMP-9 was in tissue samples from all three groups (control, placebo, and C-EGF-D IHG) on days 7,14 and 21. This was done for all groups. 

#### 4.6.3. TNF-α and Interleukin (IL)-10 Determination

According to the manufacturer’s instructions, we employed TNF- α and IL-10 ELISA kits (Koma Biotech, Seoul, Korea) to estimate TNF- α protein levels in the wound lysate. All four samples (protease and cytokine) were tested in triplicate, and the optical densities were checked against a standard curve to ensure accuracy. IL-10 and TNF-α were measured in picograms of proteins per milligram of total protein (TNF-α and IL-10). 

#### 4.6.4. Statistical Analysis

The data were conveyed as the mean ± the standard deviation. The statistical significance of the results was determined using a one-way analysis of variance (ANOVA) and Dunnett’s post hoc test, among other methods. The statistical analysis was carried out with the help of GraphPad Prism v8. Program. Significant values were those with *p* < 0.001, 0.01, and 0.05 or lower, respectively.

## Figures and Tables

**Figure 1 gels-09-00027-f001:**
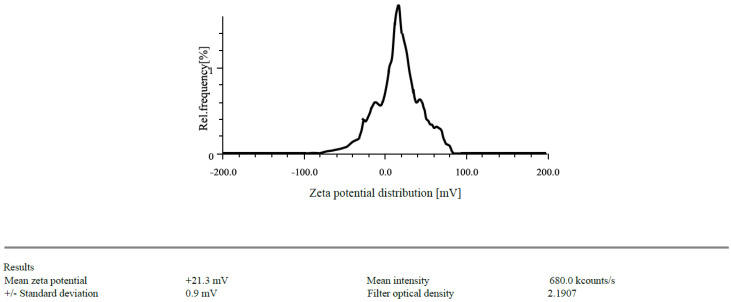
Zeta potential of CSNPs.

**Figure 2 gels-09-00027-f002:**
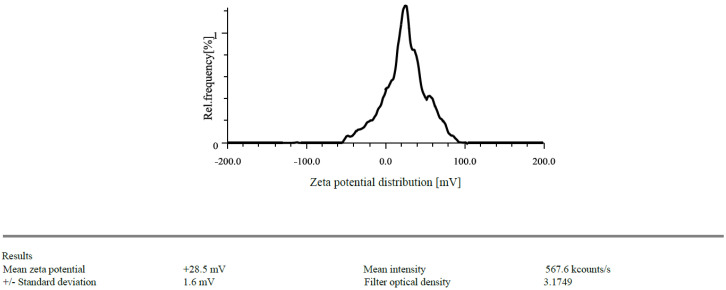
Zeta potential of EGF-CSNPs.

**Figure 3 gels-09-00027-f003:**
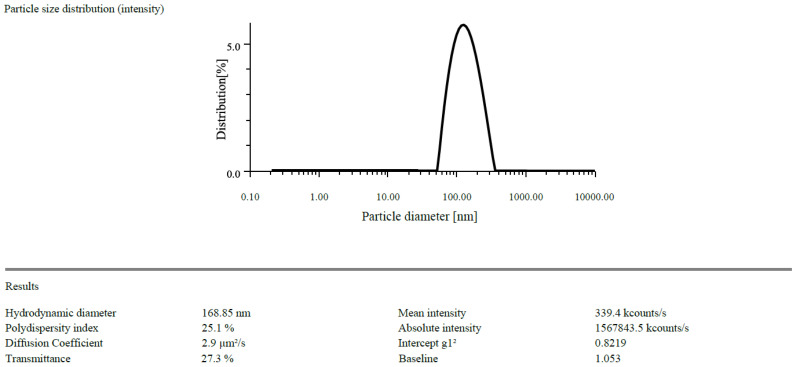
The particle size of CSNPs.

**Figure 4 gels-09-00027-f004:**
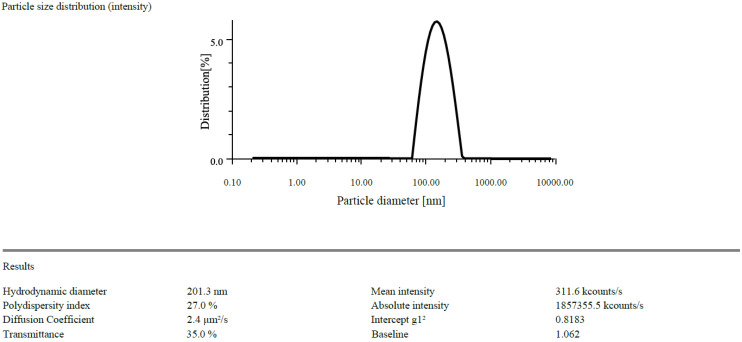
The particle size of EGF-CSNPs.

**Figure 5 gels-09-00027-f005:**
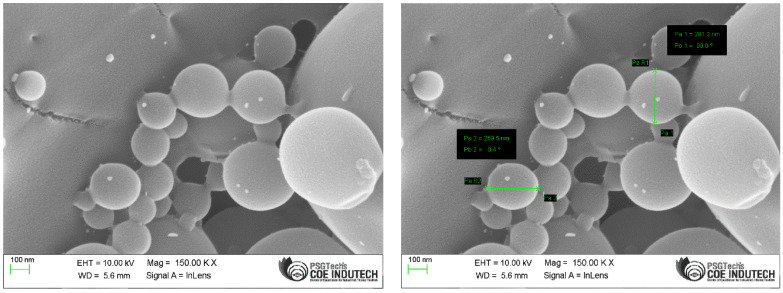
Schematic diagram of SEM images of prepared EGF-CSNPs.

**Figure 6 gels-09-00027-f006:**
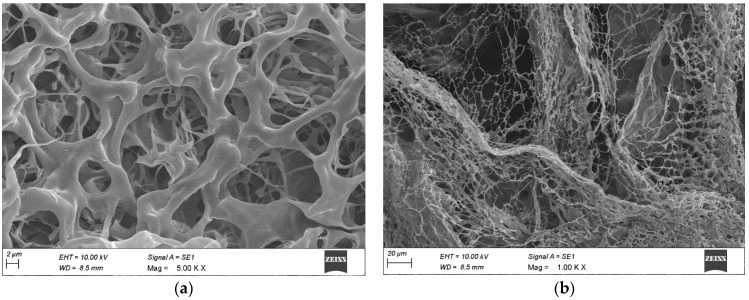
Morphology of C-EGF-D IHG determination by SEM at a scale range of (**a**) 2 and (**b**) 20 µm.

**Figure 7 gels-09-00027-f007:**
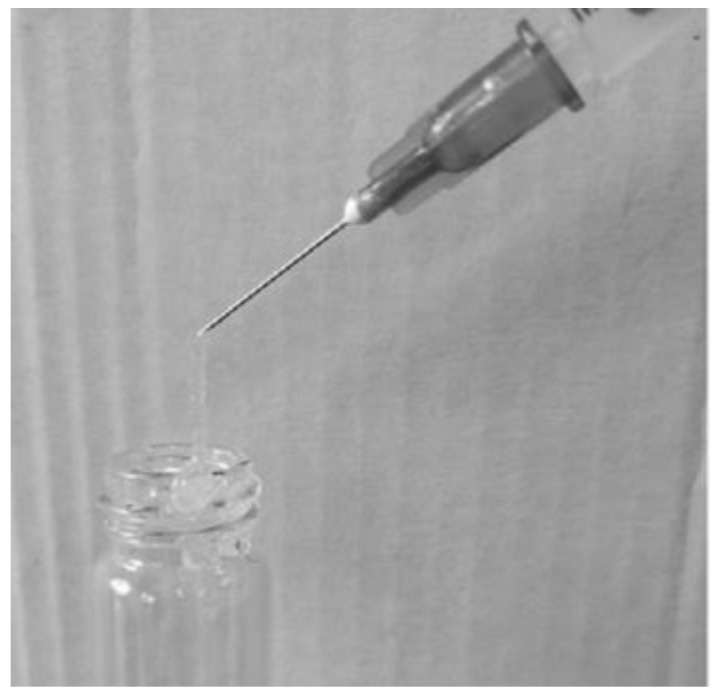
Injectability of C-EGF-D IHG.

**Figure 8 gels-09-00027-f008:**
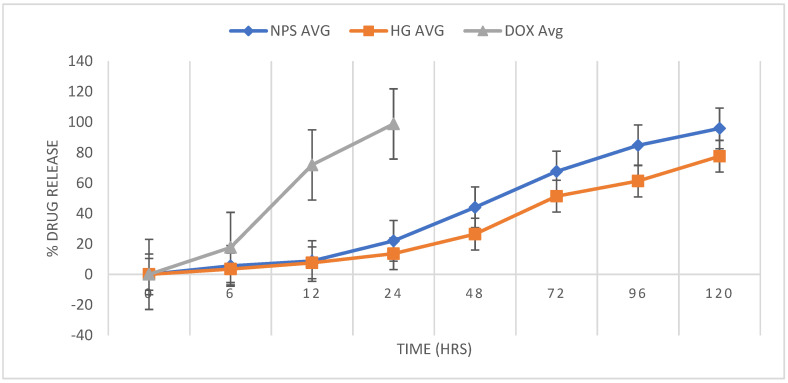
In vitro drug release profile of EGF and DOX from EGF-CSNPS and C-EGF-D IHG in phosphate buffer pH 7.4 at 37 °C.

**Figure 9 gels-09-00027-f009:**
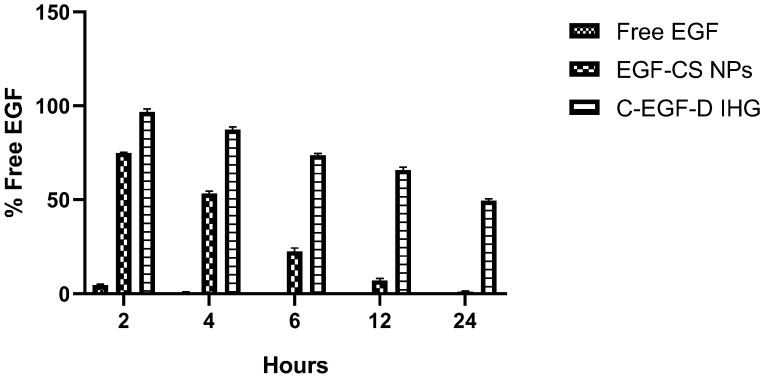
In vitro stability of EGF in the presence of savinase for 24 h.

**Figure 10 gels-09-00027-f010:**
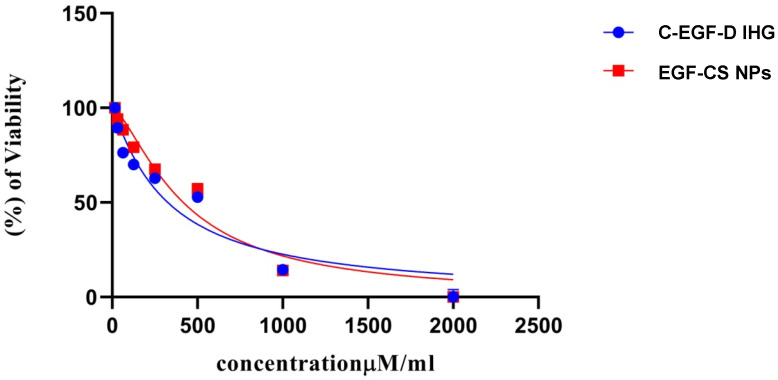
In vitro cytotoxicity studies of EGF-CS NPs and C-EGF-D IHG. *****:** significant.

**Figure 11 gels-09-00027-f011:**
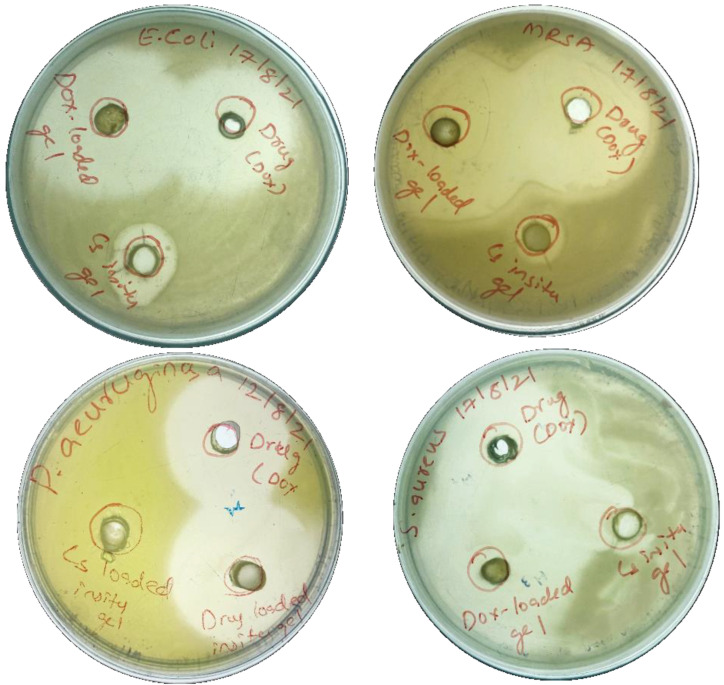
Anti-bacterial activity of DOX, CS-loaded-loaded IHG, and DOX-loaded IHG against *E. coli*, *P. aeruginosa*, *S. aureus*, *and MRSA* cells/mL (mean ± SD; n = 3).

**Figure 12 gels-09-00027-f012:**
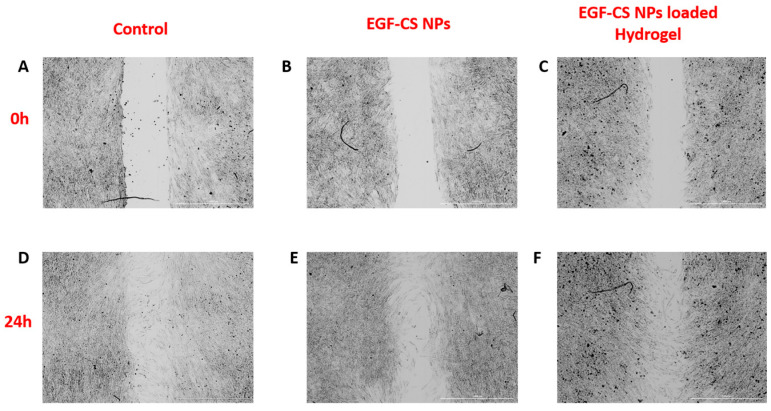
Cell migration in 3T3-L1 with or without (control) treatment for 24 h.

**Figure 13 gels-09-00027-f013:**
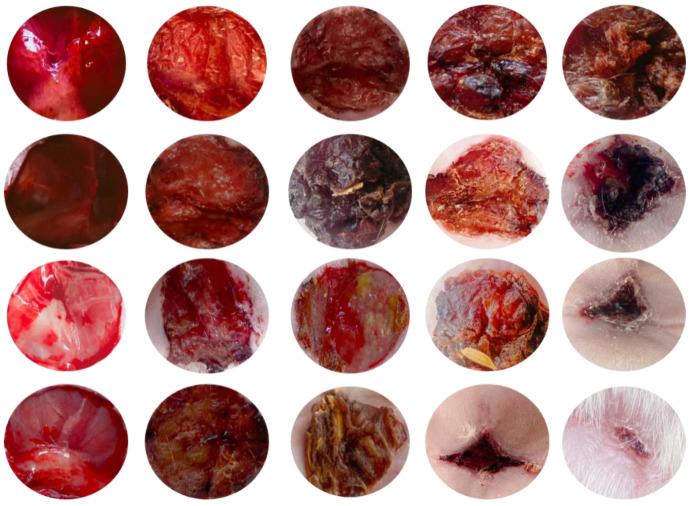
Images representing the wound contraction in control, Placebo, EGF-CS NPs, and C-EGF-D IHG treated groups from days 0 to 21 post-wounding.

**Figure 14 gels-09-00027-f014:**
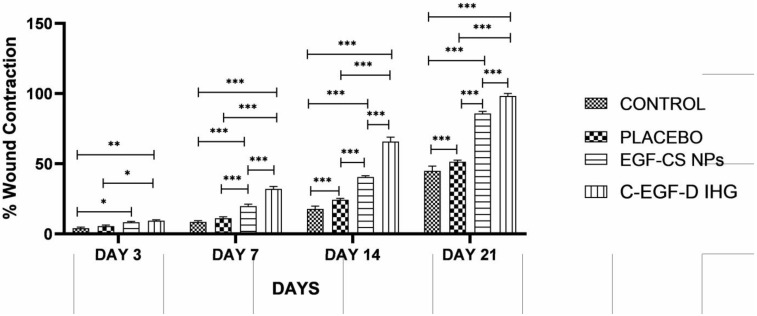
Graph representing the wound contraction in control, Placebo, EGF-CS NPs, and C-EGF-D IHG treated groups from days 0 to 21 post-wounding. * *p* < 0.05, ** *p* < 0.01 and *** *p* < 0.001 compared to control.

**Figure 15 gels-09-00027-f015:**
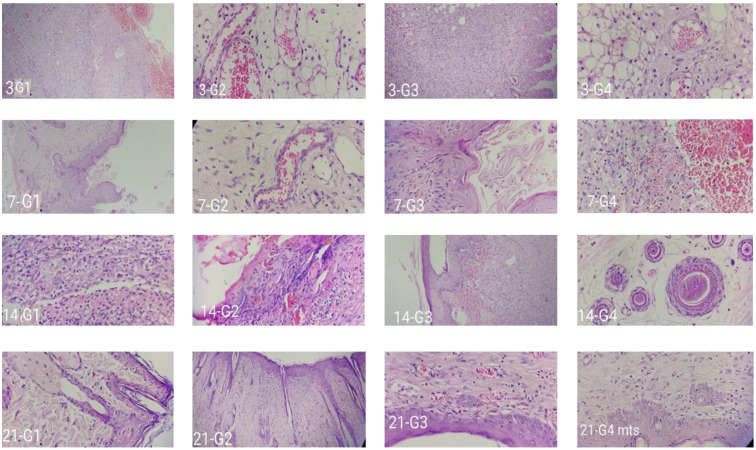
Histological changes (H & E staining) in STZ and high-fat diet-induced diabetes Wistar albino rat skin in a full-thickness excision wound model on days 7, 14, and 21 without (control) and with therapy (Placebo, EGF-CS NPs and C-EGF-D IHG.

**Figure 16 gels-09-00027-f016:**
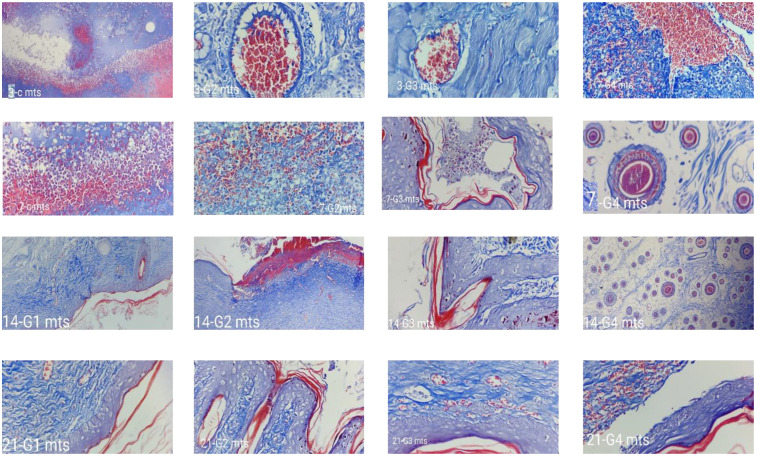
Histological changes (Masson’s trichome staining) in STZ and high-fat diet-induced diabetes Wistar albino rat skin in a full-thickness excision wound model on days 3, 7, 14, and 21 without (control) and with therapy (Placebo, EGF-CS NPs, and C-EGF-D IHG.

**Figure 17 gels-09-00027-f017:**
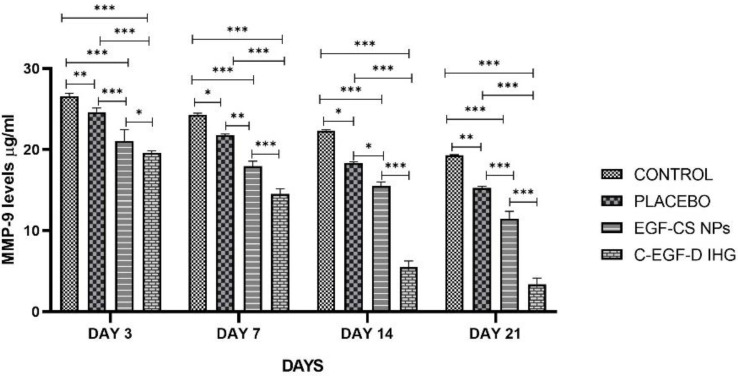
Relative protein expressions of MMP-9 level (µg/mL). * *p* < 0.05, ** *p* < 0.01 and *** *p* < 0.001 compared to control.

**Figure 18 gels-09-00027-f018:**
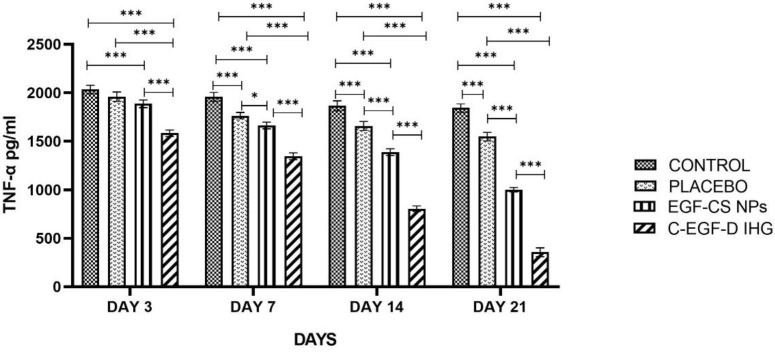
Relative protein expressions of TNF-α level (pg/mL). * *p* < 0.05 and *** *p* < 0.001 compared to control.

**Figure 19 gels-09-00027-f019:**
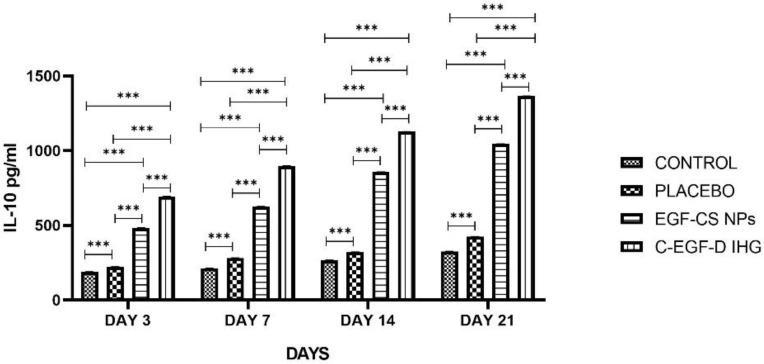
Relative protein expressions of IL-10 levels (pg/mL). *** *p* < 0.001 compared to control.

**Table 1 gels-09-00027-t001:** Formulation parameters for CSNPs.

Formulation	Chitosan	TPP	Ratio	Drug	Results
F1	0.1%	0.1%	1:1	-	No opalescence observed (no formation of particles)
F2	0.6%	0.2%	3:1	-	Size: 783 nm Zeta: +20 mV
F3	0.6%	0.1%	6:1	-	Size: 185 nm Zeta: +21.3 mV
F4	0.2%	0.8%	1:4	-	Larger particles agglomeration
F5	0.8%	0.1%	1:8	-	Size: 589 nm Zeta: +13 mV
F6	0.2%	0.4%	1:2	-	Particles not formed
F7	0.6%	0.1%	6:1	50 ug	Size: 201.3 nm Zeta: +28.5 mV

**Table 2 gels-09-00027-t002:** The Predominant bacteria observed in DW infections.

Strain	% Bacteria Isolated from DFIs
*S.aureus (Gram positive)*	38
*P.aeruginosa (Gram negative)*	17
*E.coli (Gram-negative)*	11
*MRSA* *(Gram-positive)*	05

**Table 3 gels-09-00027-t003:** Diameter of zones of inhibition of formulations against bacteria.

Sl. No.	Name of the Bacilli	Diameter of Zone of Inhibition (cms)		
		DOX (Free Drug)	DOX- IHG	CS IHG
1.	*E. coli*	3.59 ± 0.17	4.20 ± 0.3	1.61 ± 0.52
2.	*MRSA*	4.36 ± 0.37	4.76 ± 0.53	-
3.	*P. aeruginosa*	3.48 ± 0.15	4.15 ± 0.47	-
4.	*S. aureus*	3.89 ± 0.24	4.51 ± 0.36	-

## Data Availability

Not applicable.

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
