# Peer review of "Growth Factor Loaded Thermo-Responsive Injectable Hydrogel for Enhancing Diabetic Wound Healing"

_gels, 2022, doi:10.3390/gels9010027_

Round 1
Reviewer 1 Report
Remarks
1. The manuscript is compiled haphazardly and needs a distinct improvement in terms of English language, and sentence formation.
2. The title needs a rephrase defining the exact research question, terms tissue regeneration and diabetic wounds should be used judiciously and in the correct sense.
3. Use of abbreviations needs a careful check to improve the clarity of writing also.
4. citations are not in order. There is ref 34 cited in text and list shows only 22 references.
5. The authors should provide major rephrase to improve the overall clarity and flow of the manuscript.
Author Response
Dear reviewer,
Thank you for offering a thorough peer review. We have modified and rewritten sections of the manuscript to fully address the comments and we hope that this will comply with the referee’s remarks.
The following are the list of corrections that were made in the original manuscript. Our responses (RES) and the corrections made to each of the comments are listed below point by point:
Comments and Suggestions for Authors
- The manuscript is compiled haphazardly and needs a distinct improvement in terms of English language, and sentence formation.
We are very sorry for this, and the manuscript has been checked through Grammarly premium version has been updated.
- The title needs a rephrase defining the exact research question, terms tissue regeneration and diabetic wounds should be used judiciously and in the correct sense.
As suggested by the reviewer the title has been modified to define the research question.
- Use of abbreviations needs a careful check to improve the clarity of writing also.
As per the suggestion given by the reviewer, we have checked the abbreviations and abbreviated where they are appropriate.
- citations are not in order. There is ref 34 cited in text and list shows only 22 references.
We are very sorry for this improper citation and the references have been updated.
- The authors should provide major rephrase to improve the overall clarity and flow of the manuscript.
Entire manuscript has been checked by Grammarly Premium version and has been rephrased where it was found to be inappropriate.
Reviewer 2 Report
Vyshnavi Tallapaneni et al. designed and developed EGF and Doxycycline loaded hydrogel for diabetic wound healing application. This manuscript was well organized and written. Some revisions should be conducted before publication.
1. The title is a little bit confused, and it should be rewritten.
2. Abstract must be presented in a better and clear way. For instance, some important data should be presented in the Results and Conclusion part. The Background part is too long, and it should be shorten.
3. Some more descriptions should be added to introduce the recent advances of hydrogel-based wound dressings in Introduction section, and further highlight the novelty of present study.
4. The drug encapsulation rate in nanoparticles should be given.
5. Statistical analysis should be performed for all the biological tests. Do they have any significant differences in Figure 9 and Table 3?
6. Scale bars in Figure 12, Figure 15 and Figure 16 are missing, which should be added.
7. The quantitative analysis for collagen deposition should be conducted for Figure 16.
8. Some related works like 10.3390/gels8100609, 10.1016/j.apmt.2022.101542, and 10.3390/gels7040204 should be discussed to strengthen the readability.
9. For animal study, were the murine wounds splinted? This is important to mention as murine wounds are known to heal by contraction rather than re-epithelialization. These points should be mentioned and discussed in the manuscript.
Author Response
Dear sir,
Thank you for offering a thorough peer review. We have modified and rewritten sections of the manuscript to fully address the comments and we hope that this will comply with the referee’s remarks.
The following are the list of corrections that were made in the original manuscript. Our responses (RES) and the corrections made to each of the comments are listed below point by point:
Vyshnavi Tallapaneni et al. designed and developed EGF and Doxycycline loaded hydrogel for diabetic wound healing application. This manuscript was well organized and written. Some revisions should be conducted before publication.
- The title is a little bit confused, and it should be rewritten.
The title has been modified a little as per the suggestion given.
- Abstract must be presented in a better and clear way. For instance, some important data should be presented in the Results and Conclusion part. The Background part is too long, and it should be shorten.
I agree with the reviewer that the background part is too long and has been shortened to some extent. I believe that all the data has been represented already in the results and conclusion part.
- Some more descriptions should be added to introduce the recent advances of hydrogel-based wound dressings in Introduction section, and further highlight the novelty of present study.
As per the suggestions given by the reviewer recent advances of hydrogel-based wound dressings have been added and the novelty of the present study has been highlighted in the manuscript.
- The drug encapsulation rate in nanoparticles should be given.
The drug encapsulation has been already stated in the manuscript.
- Statistical analysis should be performed for all biological tests. Do they have any significant differences in Figure 9 and Table 3?
Statistical analysis has been performed in all cases as results have been done in triplicates. I believe there is a significant difference in Figure 9 and in Table 3.
- Scale bars in Figure 12, Figure 15 and Figure 16 are missing, which should be added.
Dear reviewer, we cannot include the scale bars for these figures as we are not calculating % of various parameters in the case of histopathology studies and in the case of cell migration we are noticing if there is proliferation after the treatment of the drug.
- The quantitative analysis for collagen deposition should be conducted in Figure 16.
Dear reviewer, we are sorry that have gone through many papers and could not find how to calculate % of collagen deposition and hence not included in the manuscript.
- Some related works like 10.3390/gels8100609, 10.1016/j.apmt.2022.101542, and 10.3390/gels7040204 should be discussed to strengthen the readability.
I agree with the reviewer and the studies as suggested by the reviewer have been included in the manuscript.
- For animal study, were the murine wounds splinted? This is important to mention as murine wounds are known to heal by contraction rather than re-epithelialization. These points should be mentioned and discussed in the manuscript.
Thanks for the update but the murine wounds were not splinted hence it’s not mentioned in the manuscript. In future, we can carry out the studies as per the suggestion given.
Reviewer 3 Report
The paper entitled “Growth Factor Loaded Biomimetic Thermo-responsive Injectable Hydrogel for Enhancing Tissue Regeneration 3 and Diabetic Wounds” In this manuscript the authors investigated the use of chitosan nanoparticles impregnated with thermoresponsive injectable hydrogels with protease inhibitor with EGF and doxycycline. The concept seems to be nice but still there are some suggestions to improve the quality of the manuscript.
· The encapsulation efficiency of EGF-CSNPs was up to 83.430? what does that mean? The encapsulation should be done at least in triplicates to have the encapsulation with standard deviation. Please explain. Encapsulation efficiency seems less than other delivery systems which is mostly higher than 90 percent most of the time. Please explain?
· what about the drug loading of the system.
· In the SEM analysis it looks like the particles are of variable size and it looks like the size of some of the particles looks more than what should be for injectable delivery systems. Please explain.
· The zeta potential graph looks like it has wide distribution of the zeta potential. Please explain.
· Can the author please include the data of the drug release modelling of the systems to know which kind of the kinetics is following in drug release.
· The antibacterial effect of the various formulation in comparison to the free drug looks more but is that a significant difference? Please explain in the manuscript with the data included.
· In 2.4.1 the discussion part says that, On day 7, diabetic rats treated with C-EGF-D IHG wound repair were substantially faster than EGF-CS NPs, placebo, and untreated rats, with 52.282.96 percent compared to 72.512 percent for placebo versus s94.282 percent for untreated But in the figure 14 the graph of EGF-CS NPs is higher than C-EGF-D IHG??? Not sure why please explain.
All considered I endorse this manuscript suitable for publication in Gels Journal provided all the suggestions needs to be taken care to improve the quality of the manuscript. It needs to be deeply revised before considered for the publication.
Author Response
Dear sir,
Thank you for offering a thorough peer review. We have modified and rewritten sections of the manuscript to fully address the comments and we hope that this will comply with the referee’s remarks.
The following are the list of corrections that were made in the original manuscript. Our responses (RES) and the corrections made to each of the comments are listed below point by point:
Comments and Suggestions for Authors
The paper entitled “Growth Factor Loaded Biomimetic Thermo-responsive Injectable Hydrogel for Enhancing Tissue Regeneration 3 and Diabetic Wounds” In this manuscript the authors investigated the use of chitosan nanoparticles impregnated with thermoresponsive injectable hydrogels with protease inhibitor with EGF and doxycycline. The concept seems to be nice but still there are some suggestions to improve the quality of the manuscript.
1)The encapsulation efficiency of EGF-CSNPs was up to 83.430? what does that mean? The encapsulation should be done at least in triplicates to have the encapsulation with standard deviation. Please explain. Encapsulation efficiency seems less than other delivery systems which is mostly higher than 90 percent most of the time. Please explain?
After being raised this question I also have gone through so many papers and encapsulation was in the range of 83-87 per cent and very few studies have shown around 90 percent. Encapsulation efficiency is influenced by many factors like concentration of the polymer, solubility of the polymer in a solvent, rate of solvent removal, the solubility of organic solvent in water. Literature suggests that more than 80 percent is well enough for the maximum loading of drug. Encapsulation efficiency had been performed in triplicates.
2) what about the drug loading of the system.
The drug loading is what the encapsulation efficiency refers to (defined by the concentration of the incorporated material (such as active ingredients, drugs, fragrances, proteins, pesticides, antimicrobial agents, etc.) detected in the formulation over the initial concentration used to make the formulation. It ahs been given in the manuscript under encapsulation efficiency.
3) In the SEM analysis it looks like the particles are of variable size and it looks like the size of some of the particles looks more than what should be for injectable delivery systems. Please explain.
I agree with the reviewer that there are very few particles that are of variable size and are well below the size specific for injectable drug delivery system and it has shown in the figure 7 the injectability of the formulation.
4)The zeta potential graph looks like it has wide distribution of the zeta potential. Please explain.
I am sorry that I didn’t get this question from the reviewer, and I wonder whether this influences the stability of particles.
5) The antibacterial effect of the various formulation in comparison to the free drug looks more but is that a significant difference? Please explain in the manuscript with the data included.
DOX is a potent antibacterial well proven drug but has shown synergistic effect combined with chitosan. But compared to the free drug DOX, the hydrogel containing drug and chitosan has synergistic effect and hence the study. I hope this justifies the question.
6) In 2.4.1 the discussion part says that, On day 7, diabetic rats treated with C-EGF-D IHG wound repair were substantially faster than EGF-CS NPs, placebo, and untreated rats, with 52.282.96 percent compared to 72.512 percent for placebo versus s94.282 percent for untreated But in the figure 14 the graph of EGF-CS NPs is higher than C-EGF-D IHG??? Not sure why please explain.
I am extremely sorry for the mislabelling of the content and now the changes have been done and the hydrogel has shown better healing compared to other formulations. I thank you reviewer for letting us know about this.
All considered I endorse this manuscript suitable for publication in Gels Journal provided all the suggestions needs to be taken care to improve the quality of the manuscript. It needs to be deeply revised before considered for the publication.
Round 2
Reviewer 1 Report
The authors have improved the manuscript and now seem well to be considered for publication.
Author Response
Dear Reviewer,
We thank you very much for the valuable comments and suggestions.
Reviewer 3 Report
· The Author have explained some of the raising concers about the study but still there are some more concerns or suggestions which needs to be improved.
· The author did not understand the concept of drug loading and encapsulation efficiency. Encapsulation efficiency is different from drug loading in the systems please include that data.
· Synergistic effect is a big terms to justify the answer. Did the author used any of the equation to justify that it has synergism. Please explain and include that data.
After careful considering the authors response I would recommend to include these data, can be accepted for publication to improve the quality of the publication
Author Response
The Author have explained some of the raising concers about the study but still there are some more concerns or suggestions which needs to be improved.
1) The author did not understand the concept of drug loading and encapsulation efficiency. Encapsulation efficiency is different from drug loading in the systems please include that data.
RES: Dear reviewer, we are very sorry for the mis-conceptualization. We agree with you that both terms differ from each other and have added the drug-loading data as requested.
2) Synergistic effect is a big terms to justify the answer. Did the author used any of the equation to justify that it has synergism. Please explain and include that data.
RES: The word synergism describes the interaction of two or more drugs when their combined effect is greater than the sum of the effects seen when each drug is given alone. Since we have seen the increased effect when drug has been used with chitosan than when they are used alone, we have used the word synergism in responses. However, the word synergism is not included in the manuscript and has not used any equation to calculate the same.
Thank reviewer for your valuable comments.